# STEALING THE INVISIBLE: UNVEILING PRE-TRAINED CNN MODELS THROUGH ADVERSARIAL EXAMPLES AND TIMING SIDE-CHANNELS

## ABSTRACT

Machine learning, with its myriad applications, has become an integral component of numerous technological systems. A common practice in this domain is the use of transfer learning, where a pre-trained model's architecture, readily available to the public, is fine-tuned to suit specific tasks. As Machine Learning as a Service (MLaaS) platforms increasingly use pre-trained models in their backends, it's crucial to safeguard these architectures and understand their vulnerabilities. In this work, we present an approach based on the observation that the classification patterns of *adversarial images* can be used as a means to steal the models. Furthermore, the adversarial image classifications in conjunction with *timing side channels* can lead to a model stealing method. Our approach, designed for typical user-level access in remote MLaaS environments exploits varying misclassifications of adversarial images across different models to fingerprint several renowned Convolutional Neural Network (CNN) and Vision Transformer (ViT) architectures. We utilize the profiling of remote model inference times to reduce the necessary adversarial images, subsequently decreasing the number of queries required. We have presented our results over 27 pre-trained models of different CNN and ViT architectures using CIFAR-10 dataset and demonstrate a high accuracy of $88.8\%$ while keeping the query budget under 20.

## 1 INTRODUCTION

The rapid growth of Machine Learning (ML) has transformed various industries. However, the complexity and resource intensity of developing in-house models have paved the way for Machine Learning as a Service (MLaaS) (Ribeiro et al., 2015). Companies like Google and Amazon provide businesses access to advanced, pre-trained ML/DL models via cloud services, eliminating the overhead of internal development and maintenance. However, the widespread use of MLaaS has amplified concerns around model privacy and security. These models, loaded with proprietary data and unique algorithms, are vital intellectual properties that offer competitive edge. In such a highly competitive environment, the security of these models is at risk due to the rise of techniques for reverse-engineering or "stealing" (Oliynyk et al., 2022). Increased research in model stealing poses a significant threat to the proprietary rights and market position of MLaaS providers.

A query-based attack is a common method for model stealing, where adversaries use a model's prediction Application Programming Interface (API) to recreate or "steal" (Oliynyk et al., 2022) it without direct access to its parameters or training data. Attackers generate a set of synthetic input samples or may already have access to data of similar distribution as that of the training data and send these to the model's prediction API. Further, by analyzing the predictions, thet attempt to reverse-engineer the model, often referred to as a black-box attack because the attacker has no knowledge of the model's internal workings, but can only access its input/output interface. Numerous studies have proposed attacks on diverse ML and DL models across various modalities, including text(Krishna et al., 2020; Li et al., 2023; Pal et al., 2020), images, and graphs(Wu et al., 2022; He et al., 2021; DeFazio & Ramesh, 2019; Shen et al., 2022). In particular for attacks on image modality which is the focus of this paper, there are certain works which try to steal the target model's complete architecture and parameters (Kariyappa et al., 2021; Rolnick & Kording, 2020; Roberts et al., 2019). On the other hand there are many works which create a substitute model by replicating the performance of the

original target model (da Silva et al., 2018; Kariyappa et al., 2021; Li et al., 2018; Mosafi et al., 2019; Orekondy et al., 2019; Papernot et al., 2017; Yuan et al., 2022). Notably, the success and practicality of query-based attacks hinges on the *query budget*, which limits the number of queries one can make to an ML model in a set period to manage resources and enhance security. Hence for an attacker, reducing query numbers is vital, as excessive queries can raise alarms, leading to service suspension and a thwarted attack.

In addition to query-based attacks, there exists another category of model stealing attacks that leverage side-channel or microarchitectural leakages to extract details about the model's architecture and parameters. A substantial amount of research has focused on using side-channel information in remote settings to reverse-engineer the architecture and parameters of proprietary Deep Neural Networks (DNNs). Various studies have leveraged cache-based side-channels to recreate essential architectural secrets of the target DNN during the inference phase, using the Generalized Matrix Multiply (GEMM) operation in the DNN's implementation (Hong et al., 2018; Yan et al., 2020). Cache memory access patterns have also been exploited to gain layer sequence information of CNN models and thereafter the complete architecture by utilizing LSTM-CTC model and GANs as well. (Hu et al., 2020; Liu & Srivastava, 2020). Other investigations have exploited shared GPU resources, hardware performance counters, and GPU context-switch side-channels to extract the internal DNN architecture of the target (Naghibijouybari et al., 2018; Wei et al., 2020). Additionally, some researchers have shown that it is possible to steal critical parameters of the target DNN by exploiting rowhammer fault injections on DRAM modules (Rakin et al., 2021). Timing side-channels have been utilized to both build an optimal substitute architecture for the victim and extract DL models on high-performance edge deep learning processing units (Duddu et al., 2018; Batina et al., 2019; Won et al., 2021). Few works have also leveraged side-channel information like power (Wei et al., 2018; Yoshida et al., 2020), electromagnetic emanation (Batina et al., 2019; Yu et al., 2020; Chmielewski & Weissbart, 2021), and off-chip memory access (Hua et al., 2018) to reverse engineer architectural secrets of DNNs, which require physical access to the model.

Companies that offer APIs for specialized applications, often use models fine-tuned from standard pre-trained models like AlexNet or ResNet that are available publicly. Some research works aim to identify these underlying pre-trained/teacher models in the backend of MLaaS platforms. One such work has proposed a query based model stealing attack that relies on analyzing the classification outputs of customized synthetic input images introduced to the model with a minimum requirement of atleast 100 queries for good attack accuracy (Chen et al., 2022). From a side-channel perspective, a recent study employed GPU side-channels to identify pre-trained model architectures, but this approach used `nvprof` GPU profiler, which is mostly disabled on cloud platforms providing the MLaaS services (Weiss et al., 2023). In other work, user accessible CPU and GPU memory side-channel information were exploited to perform DNN fingerprinting on CPU-GPU based edge devices, but these won't work in cloud settings where the client only gets an API response from the server and cannot access any other information about the system (Patwari et al., 2022). *Our research is the first to combine both query based as well side-channel model stealing attack methodologies to determine the pre-trained models deployed in the backend of an MLaaS platform while only possessing client or user privileges and limiting the query requirements to less than* 20 *queries.*

With high influx of works on model stealing attacks, defending machine learning models against theft has also become of paramount importance. Various techniques, such as rate limiting and incorporating noise into the output predictions, have been devised to prevent these attacks. However, these strategies have their limitations and can impact the service utility for legitimate users. An emerging technique in the field of model IP protection is watermarking or fingerprinting models (Regazzoni et al., 2021; Lederer et al., 2023). Recently many works have also utilized adversarial examples for the same (Xue et al., 2021; Szyller et al., 2021; Zhao et al., 2020). This method works by embedding unique perturbations into the model during its training phase, which can be used as identifiers. These adversarial examples - inputs that are intentionally designed to induce model errors - act as the model's unique fingerprints and can be used as markers of authenticity. However, the fact that adversarial examples can be used to fingerprint the ML models also poses the looming danger of being used as a means of model stealing.

Adversarial example show an intriguing property of transferability (Liu et al., 2017), observed in machine learning models, particularly deep neural networks (DNNs). This property means that an adversarial example, originally designed for a specific machine learning model, can also affect other models, leading to successful misclassifications. In this work, we demonstrate and emphasize on

the fact that, although adversarial examples can transfer between models, they may not necessarily be classified into the same class as the initial model due to difference in the decision boundaries of various models. We exploited these divergent misclassifications of adversarial images from different models to fingerprint several renowned pre-trained CNN architectures.

We work with assumption that the adversary does not have any knowledge about target model's architecture as well as weight parameters. For this, we utilize a window of top classifications from the MLaaS server. For each architecture we profile with multiple models of varying weight parameters to better classify the target model. *It is to be noted, our work is the first one to exploit adversarial image classification pattern among various CNN architectures to reverse-engineer CNN models.* Our work shows that an effective combination of adversarial image selection and timing based side-channels can be used to discern the target CNN models, with as little as 15 observations, thus reducing the query budget requirement for the attack.

This strategy significantly helps in reducing our query requirements, allowing us to maintain it below ten queries for a successful attack. We have shown results for our attack with the standard CI-FAR10 (Krizhevsky et al., 2009) dataset. Furthermore, we have worked with 27 pre-trained models of different CNN and ViT architectures provided by PyTorch (Paszke et al., 2019) and Hugging-Face (Wolf et al., 2019) respectively. Next, we summarize the contribution of this work:

- We observe that while transfer learning of CNN architectures allow adversarial attacks the target classes are distinct and can be used for fingerprinting.

- We present a 2-staged model stealing attack, exploiting the remote inference timing side-channels in the first stage to shortlist potential architectures and prediction pattern of adversarial images in the second stage for final prediction.

- We show through extensive experiments on 27 pretrained models available on PyTorch and Hug-gingFace using CIFAR10 dataset that our model stealing attack works accurately even in situations where the weights of the target model vary significantly compared to state-of-the-art.

## 2  RECOGNIZING ADVERSARIAL IMAGES AS ARCHITECTURE IDENTIFIERS

We are aware of the transfer-ability property inherent in adversarial examples, whereby an adversarial image generated to induce misclassification in a particular Machine Learning (ML) model may also succeed in causing misclassification when presented as input to other ML models. We emphasize that while transfer-ability implies that the misclassification extends across models the resulting misclassified class is not the same. In this section, we delve into these misclassification patterns observed among various pre-trained, or teacher models and evaluate their ability of fingerprinting these models and subsequently exploit it to orchestrate a model extraction attack on Machine Learning as a Service (MLaaS) servers. A comprehensive exploration of these adversarial examples application is discussed in Sections 4.

**Experimental Setup:**  We perform our experiments using a total 27 pre-trained image classification models including both CNN and ViT architectures. We show the list of models in Table 1, where we group the models under different architecture types. While the experiments were performed using PyTorch, it should be noted that they are not limited to this particular Deep Learning (DL) framework and can be replicated using alternative frameworks such as TensorFlow and Caffe. For the adversarial example generation we use the three well-known algorithms namely, Fast Gradient Sign Method (FGSM), Projected Gradient Descent (PGD) and Basic Iterative Method (BIM). We use CIFAR-10 dataset for model training and adversarial examples generation. The models are finetuned for these datasets over the pre-trained models provided by PyTorch, which are originally trained on Imagenet-1k dataset.

Consider a scenario where we have a set of models $M = \{M_1, M_2, \ldots, M_Z\}$. From this set, we select a single model, say $M_i$, $1 \leq i \leq Z$ as the base model to generate $N$ adversarial examples employing any of the recognized adversarial attack strategies. We then give these $N$ adversarial images as input to all the remaining $Z$ models and observe the classification pattern for each image. The primary objective is to determine whether these classifications are uniform across all models or whether they show variation. For our initial experiment, we have $Z = 27$, $M_i$ is `Resnet-18` model and we generate $N = 1000$ adversarial images of CIFAR-10 dataset using FGSM, BIM and PGD adversarial attacks. In Figure 1 we show classification for a sample of 5 adversarial images

Table 1: List of Models and their Groups

| Group Name | Models |
|---|---|
| **AlexNet** (Krizhevsky et al., 2012) | AlexNet |
| **VGG** (Simonyan & Zisserman, 2015) | VGG-11, VGG-13, VGG-16, VGG-19 |
| **ResNet** (He et al., 2016) | Resnet-18, Resnet-34, Resnet-50, Resnet-101, Resnet-152 |
| **Squeezenet** (Iandola et al., 2016) | Squeezenet1.0, Squeezenet1.1 |
| **Densenet** (Huang et al., 2017) | Densenet-121, Densenet-161, Densenet-169, Densenet-201 |
| **Inception** (Szegedy et al., 2016) | Inception v3 |
| **GoogleNet** (Szegedy et al., 2015) | GoogleNet |
| **ShuffleNet** (Ma et al., 2018) | ShuffleNet V2 |
| **MobileNet** (Sandler et al., 2018) | MobileNet V2 |
| **ResNeXt** (Xie et al., 2017) | ResNeXt-50-32x4d, ResNeXt-101-32x8d |
| **Wide ResNet** (Zagoruyko & Komodakis, 2016) | Wide ResNet-50-2, Wide ResNet-101-2 |
| **MNASNet** (Tan et al., 2019) | MNASNet 1.0 |
| **Google ViT** (Wu et al., 2020) | google/vit-base-patch16-224-in21k |
| **Microsoft Swin** (Liu et al., 2021) | microsoft/swin-base-patch4-window7-224 |

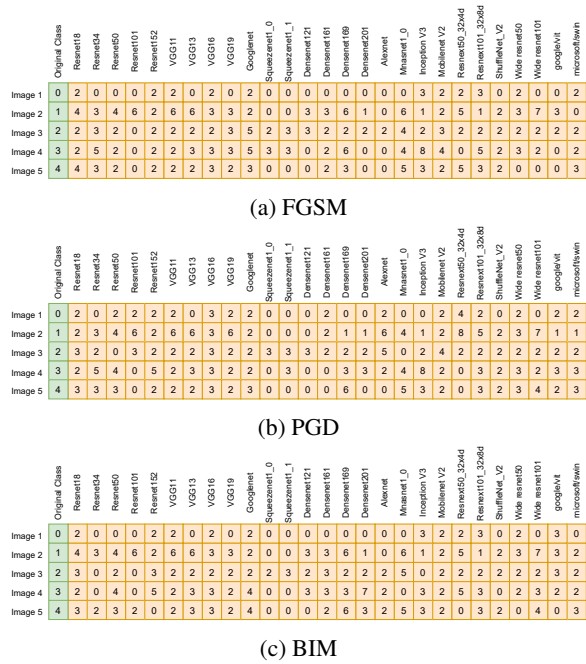

Figure 1: Varying classification for 5 adversarial images generated using FGSM, PGD, and BIM attacks, belonging to 5 different classes of CIFAR-10 dataset for 27 pre-trained models

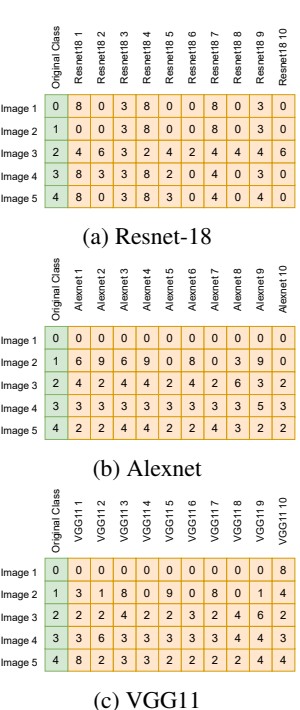

Figure 2: Classification of adversarial images generated using PGD with models of the same architectures but different weight parameters

(out of total 1000 images) from 5 different classes of CIFAR-10 dataset on 27 pre-trained CNN and ViT models. The classification for adversarial images generated using FGSM, PGD and BIM are shown in Figure 1a, Figure 1b, and Figure 1c respectively. For each adversarial image, we observe that the misclassified label by each model is not same and it varies for all the models. This trend is consistent for not only the five images depicted in Figure 1 but also for the entire set of 1000 adversarial images. Based on our observation, in Section 4 we demonstrate how we can leverage the unique misclassification trend among different pre-trained models to fingerprint them and then execute a successful model extraction attack. However, prior to that, we furnish the threat model in the following section.

## 3 THREAT MODEL

In this section we define the threat model for the proposed model extraction attack. We consider an MLaaS scenario, where a ML service provider provides API access to one of the trained ML model which has been deployed on the cloud server, to all the authorized clients. The adversary is also an authorized client of the service. The clients have no knowledge about the ML model's architecture running on the server. The adversary does not have knowledge about the target model's architecture, and further it does not have information about the weights as well.

**Adversary's capabilities:** Unlike other works (Weiss et al., 2023; Patwari et al., 2022) the adversary only has API access to the MLaaS model, through which it's impossible to use any CPU and GPU profiling tools on the server. Additionally, the adversary has only client-level privileges, and can get the execution time of each image's inference query to the model. The adversary has access to publicly available pre-trained models which he can fine-tune for particular datasets. The adversary has access to the dataset belonging to the same distribution as target model's training data.

**Adversary's Objective:** The primary objective of the adversary is to discern the pre-trained model utilized in training the target model that operates on the MLaaS server. The adversary seeks to extract the model by analyzing the classifications of the input image and its corresponding inference time, and ensuring minimum possible queries to the MLaaS.

## 4 MODEL FINGERPRINTING USING ADVERSARIAL EXAMPLES AND TIMING SIDE-CHANNEL

In this section, we extend the discussion from Section 2, where we highlighted the non-uniform classifications by pre-trained models on adversarial images. We illustrate how to identify a minimal subset of adversarial images that can effectively profile all the pre-trained models. Additionally, we demonstrate that by leveraging timing side-channels, we can further reduce the size of this minimal adversarial set. This is achieved by focusing on models whose inference time aligns closely with that of the target model operating on MLaaS server.

### 4.1 MODEL PROFILING WITH ADVERSARIAL IMAGES

In Section 2 we showed varying classification patterns for different model architectures, but model for each architecture had a specifc set of weights which in realistic scenario won't be same even though the architecture remains same. This is because various possible initial parameters which are set before training may vary for different models. Thus, we work under the assumption of a weight-oblivious adversary, implying that we are unaware of the model architecture and its weights. For such an adversary, it is essential to create profiles for numerous models with the same architecture but with differing weight parameters.

We have total of $Z$ different model architectures. We train $k$ models of each architecture with varying weights. We generate a set of $N$ adversarial images and generate classification for all $k$ models for each architecture. For our experiment, without loss of generality, we took $Z = 27$, $k = 10$ and $N = 1000$. In Figure 2, we show classification for 5 (out of 1000) adversarial images of CIFAR-10 dataset with 3 models, namely Alexnet, Resnet-18 and VGG11. It is visible from the figure that the the classification for each image is not consistent across all models of the same architecture. Furthermore, where classifications appear consistent—for example, for the class 0 image—the results are comparable across all three architectures, making it difficult to distinguish between different architectures using such images. Consequently, we chose to evaluate the top-5 classifications of the model, rather than just the top-1. This approach provided us with deeper insights into the varied classification patterns of the different models.

We show this by again taking the example of three architectures, Alexnet, Resnet-18, and VGG11. We trained 10 models of each architecture with different weight parameters. Next, for a particular adversarial image generated using a CIFAR-10 image with PGD, we collected top-5[1] classifications for all the 30 models. We then calculated class-wise probability means for each architecture. As a result, for every architecture, we had 10 mean values corresponding to the 10 CIFAR-10 classes. The final step is to calculate class-wise *difference of means (DoMs)* for each architecture pair, for

---

[1] We choose top-5 classifications as it is a common practice in MLaaS environments.

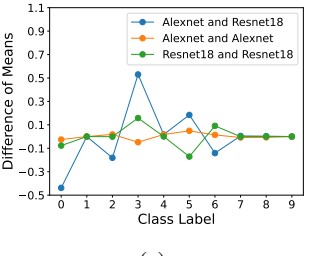 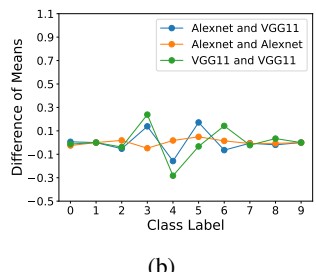 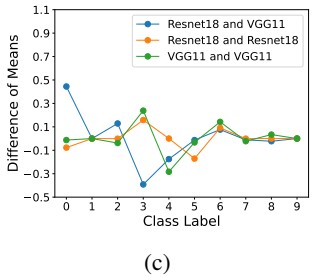

(a)                                      (b)                                      (c)

Figure 3: Comparison of Class-wise Difference of Means (DoMs) of classification probabilities between (a) Alexnet and Resnet18, (b) Alexnet and VGG11, and (c) Resnet18 and VGG11 with intra-architecture DoMs

which we have provided three plots in Figure 3. The blue line plots are for inter-architecture DoMs. Subsequently, we also show plots for intra-architecture DoMs, for which we compare first $5$ models of a given architecture with the other $5$ models of the same architecture. From Figure 3, it is evident that this specific adversarial image serves well in distinguishing between Alexnet and Resnet-18 models, as well as Resnet-18 and VGG11 models. This is observable through the high DoMs for classes 0 and 3 in both the Alexnet/Resnet-18 pair and the Resnet-18/VGG11 pair. However, for the Alexnet and VGG11 pair, there isn't a significant difference in DoMs when comparing inter and intra-architecture models.

We now formulate our methodology to discern the target model's architecture by utilizing top-5 classification information for some adversarial images. We have total of $Z$ architectures and $k$ models for all architectures with different weight parameters which were trained earlier. Let $I_x$ be an image from the adversarial image set $\{I_1, I_2, \ldots, I_N\}$. We first get classification for all $Z \times k$ models for the image $I_x$. For each model we get a vector of $5$ probabilities for top-5 class labels. Next, we transform these vectors into vectors of size $|L|$, where $L$ is the set of class-labels for any chosen dataset. In each of this vector, we place the probabilities of top-5 classes at their index values, and all other values are set to zero. With these steps our template data for all models for a particular adversarial image $I_x$ is ready. Now, for an unknown target model we pass the image $I_x$ and get the top-5 classification. We convert the result to a vector of size $|L|$ as before. The next step is to discern the architecture of the target model, by comparing the target model's generated classification vector with the prior created template. This prediction will be based on template created for one adversarial image. We select $W$ adversarial images and then create template for them. We perform majority voting on the results from template of each image. The next question that arises is how we do we decide on which images to choose for template creation and we delve into it in the next subsection.

### 4.1.1 ADVERSARIAL IMAGE SET SELECTION FOR TEMPLATE CREATION

So far, we have explored how to identify the target model by utilizing its classification of an adversarial image. The next question we address is how to determine the most effective adversarial image from a given set for this specific task. Furthermore, we need to decide the number of such images to be selected to ensure the best results with majority voting, while also optimizing this quantity to maintain the query budget.

We have a set of $Z$ potential target architectures. For each of these $Z$ architectures $k$ models have been trained with different weight parameters. Furthermore, we have top-5 classification vectors transformed into vectors of size $|L|$ for these models on a set of $N$ adversarial images. Our objective is to identify the top $d$ images that shows the highest distinguishing ability among the $Z$ architectures. To achieve this, we first compute the element-wise mean of the classification vectors from the $k$ models for each adversarial image across all architectures. This results in $Z$ vectors of size $L$ for each adversarial image. We then calculate the Euclidean distance between each pair of architecture vectors for each adversarial image and sum these distances across all pairs. The adversarial images are ranked in descending order based on this sum.

We have established a method to select the the adversarial images which can be used to distinguish between various architectures, but the number of images which we require will depend on the total number or architectures $Z$. Finally, we select the top $d$ images, those that have the highest summed Euclidean distances, as these are the images that are most effective in distinguishing the different CNN architectures. We have elaborated our methodology in a systematic manner in Algo-

**Algorithm 1** Adversarial Image Selection for Model profiling in Black-box setup

1: $N \leftarrow$ number of adversarial images
2: $Z \leftarrow$ number of architectures
3: $k \leftarrow$ number of models per architecture
4: $L \leftarrow$ size of classification vector
5: $d \leftarrow$ number of images to select
6: $ED()$: Euclidean Distance calculation
7: Initialize $V_{i,j,p}$
8: **for** $i = 1$ to $N$ **do**
9:     **for** $j = 1$ to $Z$ **do**
10:         **for** $p = 1$ to $k$ **do**
11:             $V_{i,j,p} \leftarrow \text{Classify}(i, A_j, p)$
12:         **end for**
13:     **end for**
14: **end for**
15: Initialize $D_i$
16: **for** $i = 1$ to $N$ **do**
17:     **for** $j = 1$ to $Z$ **do**
18:         $V_{i,j} \leftarrow \frac{1}{k} \sum_{p=1}^{k} V_{i,j,p}$
19:     **end for**
20:     $D_i \leftarrow \sum_{j=1}^{Z-1} \sum_{m=j+1}^{Z} ED(V_{i,j}, V_{i,m})$
21: **end for**
22: Sort $D_i$ in descending order
23: **return** top $d$ indices from $D_i$

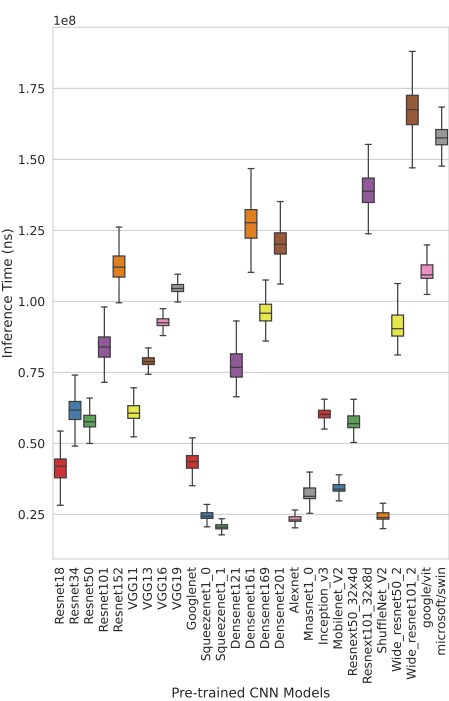

Figure 4: Box-plot for inference time distributions for 27 Pre-trained models on CIFAR-10

rithm 1. This is because to distinguish between a larger number of architectures, the requirement for the number of distinguishing images will also increase which in turn means higher query budget requirement. Also this will hamper the architecture predictability performance of our proposed approach. To address this we came up with a methodology which uses model inference times to first shortlist the potential target models, making $Z$ smaller and then applying our adversarial image selection algorithm. We discuss this in detail in the next sub-section.

## 4.2 TIMING PROFILES

The architecture of all vision models varies in terms of the number of layers, layer types, and other parameters. Primarily, the inference time of any CNN model depends on the network's depth. This is equally applicable to publicly available pre-trained CNN and ViT models. In Figure 4, we display the box-plots representing timing distributions of 27 pre-trained models, including 25 CNN from PyTorch and 2 transformer models from HuggingFace, fine-tuned on the CIFAR-10 dataset. We use `perf_counter_ns` function from the `time` Python package to collect these timings. Each timing distribution is obtained by measuring the inference time of 100 images of differing classes. Each image is processed through the model 100 times, resulting in a total of $10,000$ timing values for each distribution. The plot clearly shows that the inference times for all the 27 models vary significantly. We notice that some models have timing ranges that partially intersect with those of others. Thus, inference time alone cannot serve as a distinctive factor between the pre-trained models. However, we can use it as a criterion to filter the potential target models whose timing aligns closely with the target model's timing. Subsequently, we can utilize Algorithm 1 from Section 4.1.1 to identify the minimum set of adversarial images, which can help recognize the target model among the shortlisted ones based on inference time. It is crucial to note that by trimming down the models to create a smaller pool of possible target models, which further helps in reducing the number of adversarial images required to distinguish between shortlisted architectures. This transition further aids in lowering the query budget for the model extraction attack. In Figure 4 we observe the maximum intersection in box-plots for 6 models namely, *Resnet34, Resnet50, VGG11, VGG13, Inception-V3* and *Resnext101-32-4d* is still less than the total number of class labels in CIFAR-10 dataset.

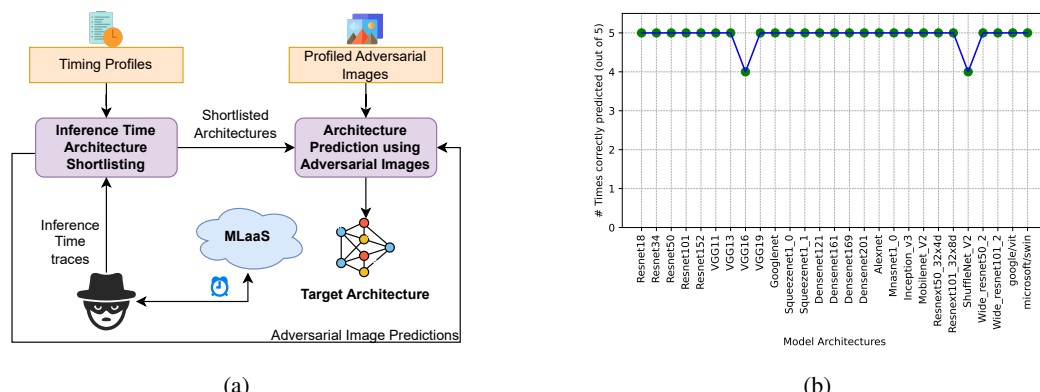

(a)                                    (b)

Figure 5: (a) Attack Methodology (b) Shortlisting correctness based on inference time for 5 test CNN target models of each architecture with varying weight parameters trained using CIFAR10.

### 4.2.1 SHORTLISTING MODELS GIVEN UNKNOWN TARGET MODEL'S INFERENCE TIME

To begin with, we will gather timing traces for all the available pre-trained models and store their maximum and minimum values range as the timing profile for each model. For every model, we will collect the inference time for different class images from the dataset, then jointly calculate their maximum and minimum range, providing a complete timing range for all image types. Let us denote the number of models as $Z$. For any model $i$, the min-max timing range is defined as $(MIN_i, MAX_i)$. We now outline the procedure for narrowing down potential target models, given the inference time of the original unknown model. Consider an unknown target model, denoted as $X$. The inference time for this model is represented as $T_X$. We use $T_X$ as a basis to select models whose min-max range encompasses $T_X$. This procedure is formally outlined in Algorithm 2 (Appendix A).

Once we have the set of potential target models, we can employ Algorithm 1 to get the minimum set of adversarial examples for the particular set of models. Then we pass the final selected images to the target model and discern it s architecture based on the model's classification outputs. The final prediction of the target model is determined through a process of majority voting. In this step, adversarial images chosen for a pre-selected group of architectures are inputted into the target model, which then yields the top-n class predictions. For every prediction made from an adversarial image, we measure the Euclidean distance to all profiled model architectures and pinpoint the one with the closest proximity. In the final phase, a comprehensive majority voting is conducted, taking into account all the models predicted by the selected adversarial images. The outcome of this collective voting determines the final prediction. The final methodology is shown in Figure 5a.

## 5 EXPERIMENTAL RESULTS AND DISCUSSION

In this section, we discuss the results for the model extraction methodologies explained in the previous section. We have performed our experiments using 27 pre-trained models including 25 CNN models provided by PyTorch via the `torchvision.models` package and 2 Vision Transformer models from HuggingFace. We further fine-tuned these models for CIFAR-10 dataset. All experiments throughout the paper have been performed on an Intel Xeon Silver 4214R CPU system with 128 GB RAM. We discuss our results with assumption of an adversary, who has no knowledge about the architecture as well as weight parameters of the target model. In this scenario, we would require multiple models to profile any particular architecture, and hence for each architecture we trained total of 15 models with varying weights for each architecture using the CIFAR-10 dataset. Among these 15 models we used 10 models for fingerprinting each architecture whereas we kept the 5 other models for testing purpose. It is to be noted that all the models have been fine-tuned on the pre-trained models, which means that the weights have been modified in all the layers of the model and not just the last layers as assumed in the prior work (Chen et al., 2022). The initial step involves profiling the inference time for each model across all architectures. We access all the model architectures remotely over an API call using the FLASK API setup. We discuss the results in detail in the next subsection.

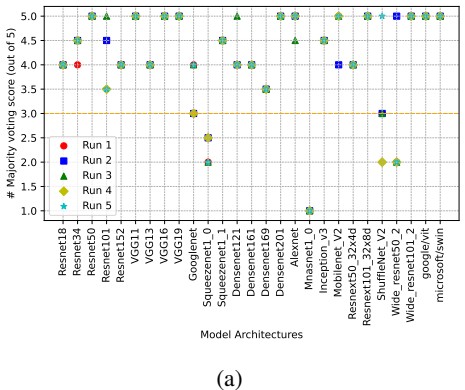 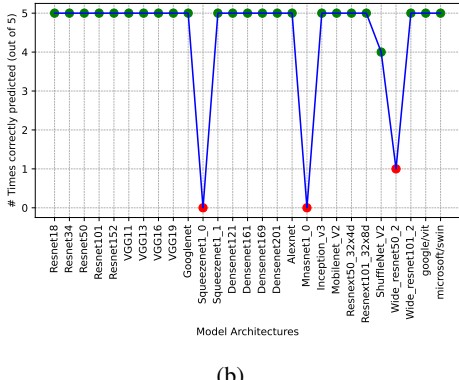

(a)                                       (b)

Figure 6: (a) Majority voting score for different architectures across 5 prediction runs. (b) Number of correctly predicted test models of different CNN architectures using CIFAR10 dataset

## 5.1 MAPPING INFERENCE TIME TO TIMING PROFILES

For every architecture, we record the inference times remotely through the API calls for all the trained models within that architecture and consolidate them. For our experiments, we have collected 10 timing traces for each model, thus accumulating a total of 100 timing values per architecture. For the target model, we collect 10 timing traces and then use Algorithm 2 to narrow down potential target models. To check the reliability of our approach we collected the timing traces for each of the 5 models of each architecture set apart earlier. In Figure 5b we show the number of times the actual target architecture got shortlisted for all the test models. We observe that out of total 27 architectures, 25 architectures are shortlisted with 100% accuracy, whereas 2 architectures, VGG-16 and Shufflenet_V2 are correctly clasified 4 out of 5 times. Overall 98.5% models are correctly shortlisted based on inference time. Now we move on to the next subsection, where we discern the target model from the shortlisted models using specifically chosen adversarial images.

## 5.2 ADVERSARIAL IMAGE SET SELECTION

We'll now employ the approach defined in Section 4.1.1 to select the images which best help in distinguishing the group of shortlisted architectures for each of the 27 target architectures. We select these images utilizing adversarial image classifications from 10 models of varying weights from each architecture. Subsequently, we check the reliability of this approach using the test models set apart earlier for each architecture. We first select top 5 adversarial images suitable for distinguishing the shortlisted architectures using Algorithm 1. Then we apply the majority voting to get the final target model prediction. In Figure 6a, we show the majority voting scores for 5 test models of each architecture over different runs. Finally in Figure 6b we show the number of correctly classified models out of the 5 test models for each target architecture after the majority voting. We observe that out of total 27 architectures 24 of them show 100% correct prediction for the 5 test models, whereas for Wide_resnet101_2 there are 4 correct predictions. Overall, we tested for $135 = 27 * 5$ models of different architecture, and we get an average accuracy of 88.8% and maximum accuracy of 92.59% for correct predictions across different runs.

## 6 CONCLUSION

In this study, we delved into the intriguing property of adversarial examples in machine learning models, with a focus on CNNs. We discovered that adversarial examples could influence the classification of various models, but don't always trigger the same misclassification due to differing decision boundaries. Utilizing this, we developed a unique fingerprinting method for renowned pre-trained CNN and ViT architectures. Furthermore, we employed timing side-channels to minimize the number of adversarial image queries required for identifying the target model. This approach greatly reduced the queries needed, typically to fewer than 20. Moreover, we demonstrate that, despite fine-tuning all layers of the pre-trained model, we successfully beat the state-of-the-art work on model fingerprinting by correctly classifying 88.8% of models from varying architectures correctly.

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

## A    ALGORITHMS

---

**Algorithm 2** Model Shortlisting Based on Inference Time

---

1: Set of models $\mathcal{M} = \{1, 2, \ldots, Z\}$
2: **for** $i = 1$ to $Z$ **do**
3:     Define $(MIN_i, MAX_i)$ for each model $i$
4: **end for**
5: Given a target model $X$ with inference time $T_X$
6: Initialize an empty set of selected models $S = \{\}$
7: **for** $i = 1$ to $Z$ **do**
8:     **if** $MIN_i \leq T_X \leq MAX_i$ **then**
9:         Add model $i$ to the set of selected models $S$
10:     **end if**
11: **end for**
12: **return** Set of selected models $S$

---

