# OpenReview forum: "Stealing the Invisible: Unveiling Pre-Trained CNN Models through Adversarial Examples and Timing Side-Channels"
_ICLR.cc/2024/Conference — ICLR 2024 Conference Withdrawn Submission_

### Official Review · Reviewer_1hVo · 2023-10-13

**Soundness:** 1 poor
**Presentation:** 2 fair
**Contribution:** 2 fair
**Rating:** 3
**Confidence:** 4

**Summary:**

In this paper, the authors combine model extraction attacks and side-channel attacks to improve the correctness and decrease query budgets of determining the victim model's structure.

**Strengths:**

1. The authors consider different model structures, including CNN and transformers.

2. The threat model is practical, in which the adversary can only call the provided APIs.

3. This proposed method is simple and easy to follow.

**Weaknesses:**

Personally, I think there are several severe flaws in this paper.

1. The experiments are all on CIFAR-10, which is a toy dataset, only containing 10 different classes. It is doubtful whether the observation of the transferability still exists on larger datasets, like ImageNet-1K.

2. As Figure 3 indicates, for some model structures, there are no significant differences across different structures and parameters. And even for models with the same structures, there still exist differences among class labels. Therefore, it is questionable whether the proposed method is reasonable.

3. For timing profiles, in Figure 4, it is clear that the inference time is short. Therefore, a very small latency from the ML service will significantly change the results. In real cases, the service will deal with millions or billions of requests, which is to say that the time latency caused by occupation is usual. Therefore, the inference time will not be a reliable measurement to determine the model structure.

4. On the other hand, the service will adopt various accelerating methods to optimize the inference process, like pruning and quantizing, which will significantly decrease the inference time. However, based on the threat model, the adversary will have no information for such optimizations. It is to say that the timing profiles cannot provide any references for the attack.

Minor:

a. The explanation of Figure 3 is not clear.

b. No ablation study of the query budgets.

c. Single dataset, CIFAR10.

**Questions:**

Please see weaknesses.

---

### Official Review · Reviewer_5A6W · 2023-10-27

**Soundness:** 2 fair
**Presentation:** 3 good
**Contribution:** 1 poor
**Rating:** 1
**Confidence:** 4

**Summary:**

The approach proposed in this work tries to infer the base pre-trained model deployed, after being fine-tuned, to perform image classification in an MLaaS scenario. To do so, the attacker compares the inference time and the transferability of a selected set of adversarial examples between a set of publicly available models and the target model.

**Strengths:**

The paper addresses an important topic, as the knowledge of the deployed pre-trained model architecture is valuable for an attacker.

**Weaknesses:**

The paper presents several issues:

1. limited attack scenario: the authors assume that the attacker knows the set of pre-trained models that might be used by the victim, and also the dataset on which they are trained. These are not strong and realistic assumptions and limit the impact of the proposed approach;

2. optimistic assumption on the inference time profiling: the main application scenario of the attack is against MLaaS platforms, therefore time measurements (which always present some randomness) are affected by the network's delay and jitter. As the inference times of considered models are in the order of nanoseconds, it is very likely that measurements are unusable;

3. missing comparison with competing approaches: the authors state that they "successfully beat the state-of-the-art work on model fingerprinting", but actually a comparison is missing. The attack should be compared at least with Chen et al., 2022, using the same experimental setting;

4. limited experimental setting: the authors only consider one dataset for the model pre-training, and one for the fine-tuning phase;

5. missing details required to reproduce the results: for instance, the number of fine-tuning epochs and the adversarial attacks parameters are not specified;

6. the limitations of the work are not discussed at all.

**Questions:**

- Could you simulate in the experiments the presence of an internet connection, in order to estimate how the attack would perform in a realistic setting?

- What happens if the pre-trained model is not among the set known by the attacker - or if it is trained on a different dataset?

- Can you please provide the complete details to reproduce the experimental results, and explicitly address the limitations of the work?

---

### Official Review · Reviewer_rKcg · 2023-10-28

**Soundness:** 3 good
**Presentation:** 3 good
**Contribution:** 2 fair
**Rating:** 3
**Confidence:** 4

**Summary:**

The paper proposes a new approach that combines query-based and side-channel model attacks to infer the architecture of a model behind an API.
Using missclassified adversarial examples to fingerprint pre-trained model architectures, the approach is able to identify which pre-trained model architecture was used. The time-side-channel attack is used to reduce the number of possible adversarial examples that need to be used for fingerprinting possible models. The approach is evaluated using 27 different models which were fine-tuned on the CIFAR-10 dataset.

**Strengths:**

- the paper is well written and relatively easy to follow
- even though for the specified setting a lot of assumptions are made, the topic is interesting and important

**Weaknesses:**

- The experiments are only conducted on the CIFAR-10 dataset. A dataset with a higher resolution would be more interesting.
- In the abstract the paper claims that these adversarial examples used to fingerprint a model can be used for a model extraction attack. However, if we only get the information which pre-trained model was used for fine-tuning, this is not a model extraction attack in my understanding.
- The adversarial examples are generated using the same dataset which is also used for training. This is a very unrealistic and a very strong assumption that the attacker has access to the same data distribution as the provider of the model. In reality the attacker doesn't even know the data distribution used for training and certainly doesn't have access to it.
- The comprehension of fig. 1 and 2 could be improved if each predicted class has its own color. This would highlight the difference between the predictions.
- To be honest I am skeptical that this approach is working if the model is fine-tuned for longer periods of time and on a harder dataset than CIFAR-10. More thorough experiments on higher resolution datasets with more classes and longer fine-tuning period would be more convincing.

**Questions:**

- Q1: As far as I can tell, the attacker can only discern the model architectures if he has a list of possible architectures. What if he has no information of possible architectures or the architecture is a custom one?
- Q2: Doesn't the timing side-channel attack depend on the hardware used by the MLaaS provider? If the attacker has different hardware, he cannot profile the timings. Would this approach then still work?

---

### Official Review · Reviewer_QATX · 2023-11-03

**Soundness:** 2 fair
**Presentation:** 2 fair
**Contribution:** 2 fair
**Rating:** 3
**Confidence:** 4

**Summary:**

This paper introduces a model extraction attack that queries the MLaaS system and, at the same time, exploits the timing side channel. Except for the profiling stage, there are two stages in the attack. In the first stage, the timing side channel is used to shortlist the candidates from a total of 27 pre-trained models. In the second stage, adversarial examples are fed into the candidates, and then outputs from candidate models are used to differentiate different architectures. The main advantage of the proposed approach is that only limited query time is needed.

**Strengths:**

- Exploiting the inference time as a side channel has an advantage that the adversary has only, as described by the authors, user-level access to the MLaaS service. This can be further exploited and combined to assist other model extraction approaches.

**Weaknesses:**

- The **necessity of adversarial examples**. One claimed contribution is to leverage adversarial examples to differentiate different models. The provided evidence only showcases, but experimental results must be provided to further support this claim. Adversarial examples transfer across different architecture. Clean samples or randomly generated samples may also transfer across different architectures.  One possible experiment is to test the performance of random noises or clean images on differentiating the models. The classification patterns on random noises would be interesting and can provide evidence on the necessity of using adversarial examples.

- Discussions on **possible defenses** is missing. For example, similar to cryptographic hardware, secure implementations, e.g., implementations where the inference time for different models is the same, could be a good defense, especially under the threat model that the adversary can only access user-level APIs. Such experiments need to be provided for completeness.

- Discussion on **commercial MLaaS APIs** would also be interesting. It would be interesting if the authors could provide details when mounting such an attack on real-world commercial MLaaS services, showing the possibilities and the limitations. It would be a valuable discussion for practitioners.

- About the **profiling budget**. Unlike hardware devices, the adversary can access an exact copy if the device is available for sale. The profiling would be hard to implement since the adversary has limited access to the APIs. Similar to template attacks, details on the profiling budget and the adversary's capability in the profiling stage can be discussed in the threat model part. In addition, In the current setup, only 27 models are used. Does it indicate that the proposed attack can only work for a limited number of models that are completely profiled in advance?

- For results in Figure 6 b, the timing profiles can already differentiate the squeezenet1_0 from most architectures. However, instead, squeezenet1_0 can not be correctly predicted in the second stage. Please clarify.

**Questions:**

See weaknesses part